# One-Dimensional Tubular Carbon Nitride Embedded in Ni$_2$P for Enhanced Photocatalytic Activity of H$_2$ Evolution

Chenyong Jiang [1], Yiwei Jiao [1], Fada Li [2], Cheng Fang [3], Jing Ding [1,*], Hui Wan [1], Ping Zhang [2] and Guofeng Guan [1,*]

[1] State Key Laboratory of Materials-Oriented Chemical Engineering, College of Chemical Engineering, Jiangsu National Synergetic Innovation Center for Advanced Materials, Nanjing Tech University, Nanjing 210009, China; jiang.cy@njtech.edu.cn (C.J.); xiongchaonjtech@163.com (Y.J.)

[2] Jiangsu Province Engineering Research Center of Visible Light Catalytic Materials Lianyungang Technical College, Lianyungang 222000, China; resinli@sina.com (F.L.)

[3] College of Chemical Engineering, Nanjing Forestry University, Nanjing 210037, China

\* Correspondence: jding@njtech.edu.cn (J.D.); guangf@njtech.edu.cn (G.G.); Tel.: +86-151-0516-4625 (J.D.); Fax: +86-025-83587198 (J.D.)

**Abstract:** Graphitic carbon nitride is considered as an ideal semiconductor material for photocatalytic hydrogen evolution due to its suitable energy band structure, durability and environmental friendliness. To further improve the catalytic performance of g-C$_3$N$_4$, nickel phosphide-loaded one-dimensional tubular carbon nitride (Ni$_2$P/TCN) was prepared by thermal polymerization and photo deposition. The beneficial effect of the one-dimensional tubular structure on hydrogen generation was mainly attributed to its larger specific surface area (increased light absorption) as well as the linear movement of the carriers, which reduced their diffusion distance to the surface and facilitated the separation of photogenerated carriers. The loading of Ni$_2$P co-catalyst improved the visible light utilization efficiency and enabled the migration of photogenerated electrons towards Ni$_2$P, which ultimately reacted with the enhanced adsorbed H$^+$ on the Ni$_2$P surface to facilitate the photocatalytic hydrogen evolution process. This study provides new clues for the further development of efficient, environmentally friendly and low-cost g-C$_3$N$_4$ catalysts.

**Keywords:** carbon nitride; photocatalytic hydrogen evolution; nickel phosphide; one-dimensional tubular





## 1. Introduction

Since the beginning of the industrial revolution, the overconsumption of fossil fuels and the excessive emission of carbon dioxide have led to problems such as energy shortages and environmental pollution, which have seriously affected the survival and development of human beings [1,2]. Hydrogen is an environmentally friendly, high-value green fuel, and the use of solar energy for the photocatalytic decomposition of water for hydrogen evolution has great potential [3]. In the past few decades, a wide variety of semiconductor catalysts have been applied to the photocatalytic hydrogen evolution, including CdS [4,5], MOFs [6,7], TiO$_2$ [8–10], ZnO [11], and Mxenes [12], etc. Graphitic carbon nitride (g-C$_3$N$_4$), as a nonmetallic semiconductor material, is considered a promising catalyst for photocatalytic hydrogen evolution reactions due to its simple preparation, low cost and suitable energy band structure (2.7 eV). However, bulk carbon nitride has the disadvantages of small specific surface area, narrow photoresponse range, and easy complexation of photogenerated carriers, which seriously hinder the photocatalytic performance of carbon nitride [13,14].

g-C$_3$N$_4$ has been intensively studied and explored by researchers around the world, and diverse approaches have been adopted to compensate for its inherent deficiencies. The modification methods of carbon nitride mainly include elemental doping [15–18], morphology modification [19–23], defect modification [24–28], and construction of heterojunctions [29–33]. The photocatalytic hydrogen evolution reaction relies on photogenerated

electrons on the surface of the photocatalyst for the reduction of $H^+$ and thus the generation of hydrogen [34]. Therefore, by modifying the morphology of carbon nitride, the specific surface area can be increased to obtain better light absorption and optimize photogenerated carrier migration paths, which ultimately facilitates the generation of hydrogen [35]. One-dimensional carbon nitride has been widely studied in the field of photocatalytic hydrogen evolution due to its excellent light absorption and carrier separation efficiency [36–39]. In recent years, transition metal phosphides have attracted much attention due to their excellent electrical conductivity and catalytic properties comparable to those of noble metals, and they have been proven able to effectively improve the performance of photocatalytic reactions [40–42]. Among them, $Ni_2P$ has been widely investigated due to its excellent charge separation efficiency and highly efficient and stable photocatalytic hydrogen precipitation [43,44]. The loading of a $Ni_2P$ co-catalyst on CN can promote the migration of photogenerated electrons from CN to $Ni_2P$, achieving efficient separation of photogenerated carriers and ultimately improving photocatalytic hydrogen production performance. Therefore, in this study, the remarkable advantages of phosphide electron co-catalysts and morphology modulation are considered. The study of the synergistic effect of $Ni_2P$ co-catalysts and one-dimensional tubular carbon nitride is necessary for improving the hydrogen evolution activity of composite photocatalysts.

Here, we prepared one-dimensional tubular carbon nitride by thermal polymerization of urea and melamine and loaded nickel phosphide co-catalysts onto the one-dimensional tubular structures using the photodeposition method. Light absorption, charge migration, and photocatalytic hydrogen production properties were analyzed. The synthesized one-dimensional tubular composite photocatalysts exhibited excellent light absorption and photogenerated carrier separation efficiency. As a result, the one-dimensional tubular loaded $Ni_2P$ photocatalysts exhibited excellent hydrogen production performance. This study provides insight into the synergistic effect of photocatalyst morphology and composite structure.

## 2. Results and Discussion

### 2.1. Catalyst Characterization

The crystal phases of the prepared catalysts were examined by XRD measurements. As shown in Figure 1, there were two peaks at 13.0° and 27.8° for pure CN, TCN and 3 wt.% $Ni_2P$/TCN composites. The peak at 13.0° corresponded to the (100) crystallographic plane with a spacing of 0.675 nm originating from the planar periodic stacking of the heptazine ring units, while the peak at 27.8° corresponded to the (002) crystallographic plane of $g$-$C_3N_4$ with a planar spacing of 0.33 nm originating from the c-axial interlayer stacking of the aromatic heterocyclic units [45,46]. Compared with pure TCN, the position of the diffraction peaks did not change after loading $Ni_2P$, indicating that $Ni_2P$ had no effect on the crystal structure of TCN. Due to the low loading of $Ni_2P$ and the good dispersion of $Ni_2P$ on the TCN surface, no diffraction peaks associated with $Ni_2P$ were found.

FT-IR spectroscopic tests were performed to better obtain information about the functional groups of the catalysts. FT-IR spectra of the tested CN, TCN and 3 wt.% $Ni_2P$/TCN composites further confirmed the similar molecular compositions among the catalysts (Figure 2). All catalysts showed a strong absorption band at 810 $cm^{-1}$, where the peak corresponded to the respiratory vibration of the heptazine unit. The peaks at 1200–1800 $cm^{-1}$ and 3000–3300 $cm^{-1}$ corresponded to the stretching vibration and N-H stretching vibration of aromatic heterocycles, respectively [26,47]. It was shown that loading $Ni_2P$ on carbon nitride did not change the original structure of carbon nitride.

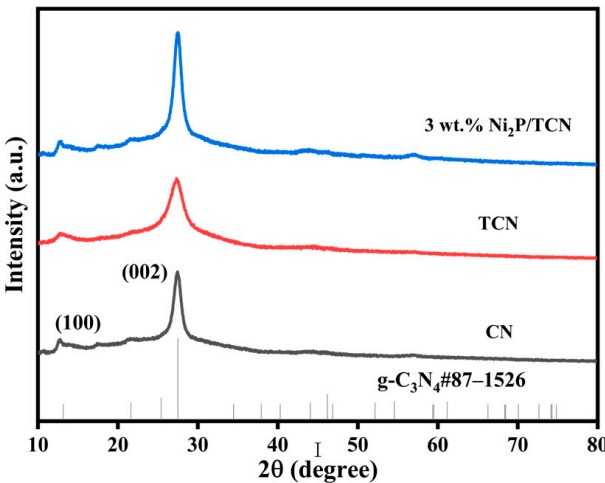

**Figure 1.** XRD patterns of pure CN, TCN, and 3 wt.% $Ni_2P/TCN$.

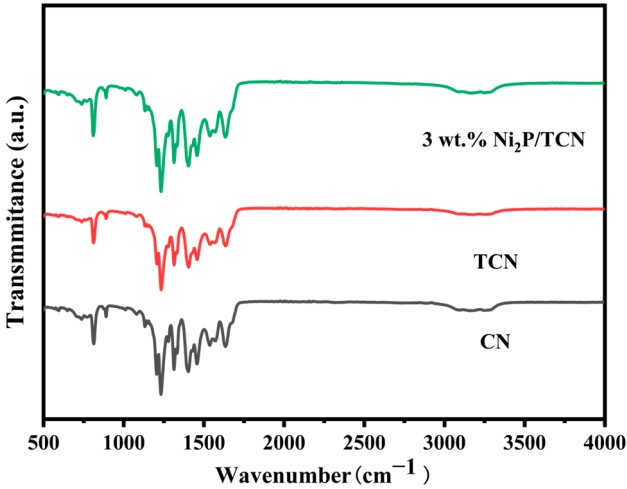

**Figure 2.** FT-IR spectra of CN, TCN, and 3 wt.% $Ni_2P/TCN$.

The micromorphological structures of CN, TCN and 3 wt.% $Ni_2P/TCN$ catalysts were investigated using scanning electron microscopy (SEM) and transmission electron microscopy (TEM). As shown in Figure 3a,b, compared with CN, TCN exhibited an obvious porous nanotube structure, and the diameters of the nanotubes were about 200–400 nm. As shown in Figure 3c–h, after loading $Ni_2P$ on TCN, TCN still retained its nanotube structure well and possessed a good hollow structure. $Ni_2P$ was uniformly dispersed in TCN nanotubes in the form of black dots. The size of $Ni_2P$ ranged from 10 to 60 nm, indicating that $Ni_2P$ existed in the form of clusters on the surface of TCN, and that the $Ni_2P$ on the TCN surface with uniform size and good distribution. From Figure 3j, it could be seen that the $Ni_2P$ NPs had clear lattice stripes, and the lattice distance of $Ni_2P$ was 0.203 nm, which belonged to the (201) crystal surface of $Ni_2P$.

In general, the activity of a catalyst strongly depended on its $S_{BET}$ value, and a larger $S_{BET}$ improved the light utilization efficiency and also promoted the photogenerated carriers transfer. The $S_{BET}$ and pore structure of pure CN, TCN and 3 wt.% $Ni_2P/TCN$ were investigated by $N_2$ adsorption–desorption isotherm measurements. As shown in Figure 4, they all have type IV isotherms and H3 hysteresis return lines (0.8 < p/p0 < 1.0), indicating the presence of interparticle mesopores. The catalyst pore size distribution was in the range of 2–60 nm. As can be seen from Table 1, the $S_{BET}$ values of pure CN, TCN, and 3 wt.% $Ni_2P/TCN$ were 74, 95, and 88 $m^2 \cdot g^{-1}$, respectively, with total pore volumes of 0.39, 0.50 and 0.42 $cm^3 \cdot g^{-1}$ and average pore diameters of 22.74, 21.80, and 19.46 nm,

respectively. The specific surface area of TCN increased by 28% as compared with that of CN. This was due to the fact that the inner and outer surfaces of the tubular structure could provide a larger specific surface area, which could provide more active sites for the reaction and promote light absorption. The 3 wt.% $Ni_2P/TCN$'s specific surface area, total pore volume, and pore size decreased compared to TCN, which was attributed to the uniform distribution of $Ni_2P$ filling the hollow structure and surface pores of TCN.

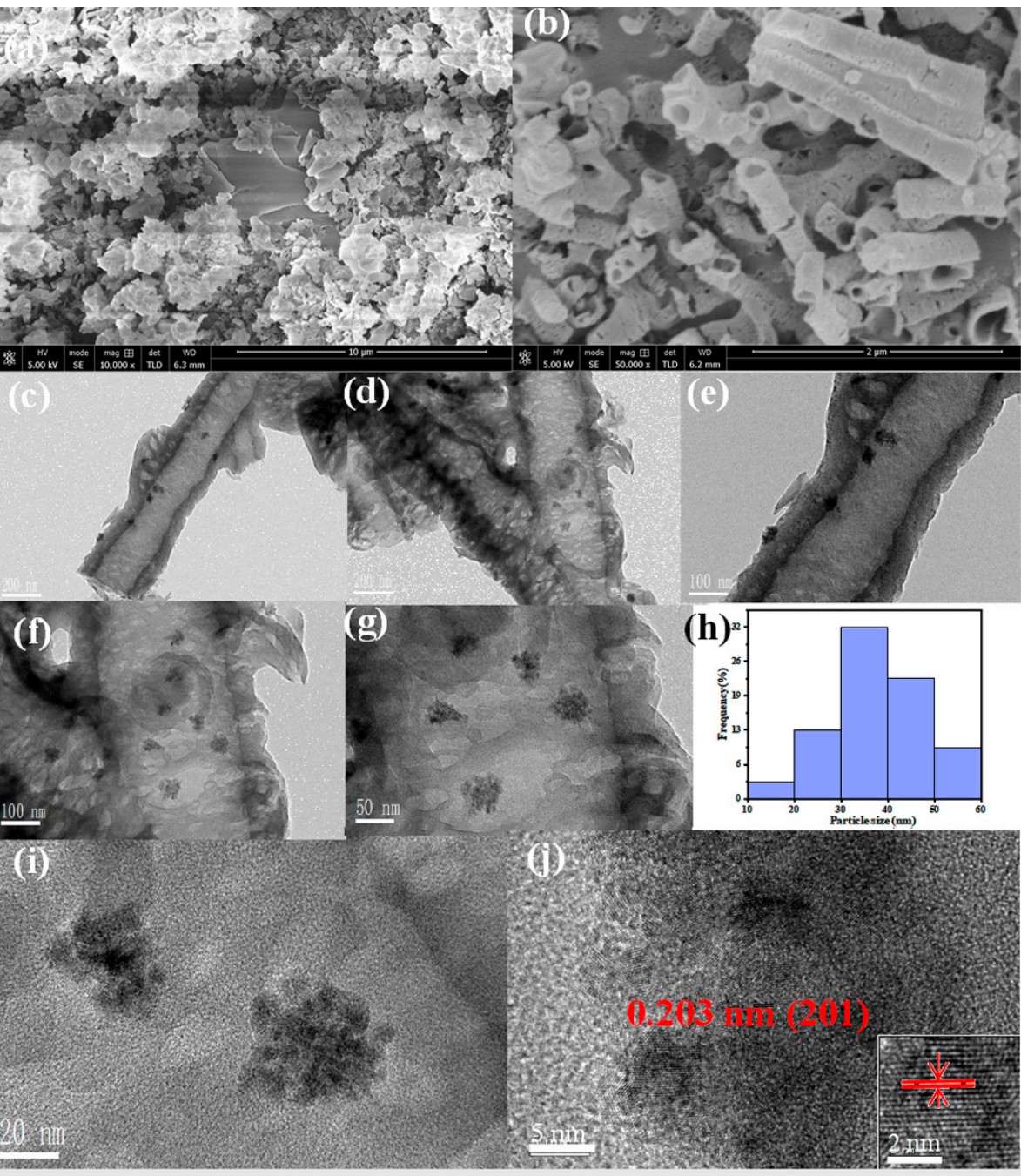

**Figure 3.** (**a**,**b**) SEM images of CN and TCN; (**c**–**j**) TEM images of 3 wt.% $Ni_2P/TCN$; (**h**) $Ni_2P$ particle size distribution.

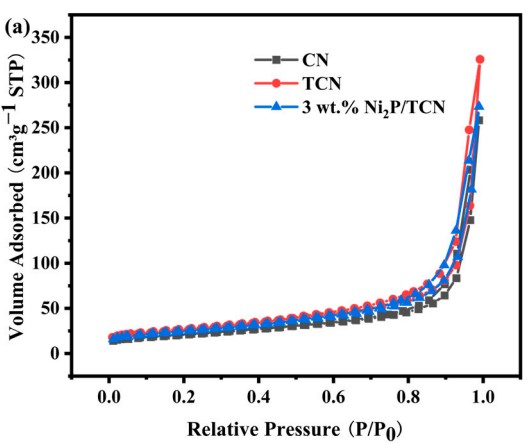
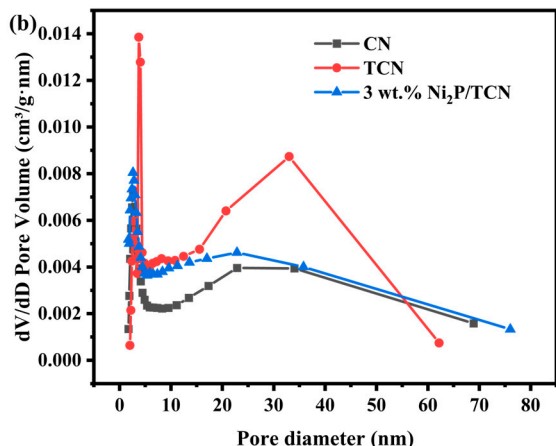

**Figure 4.** (**a**) $N_2$ adsorption–desorption curve and (**b**) pore size distribution diagram of CN, TCN, and 3 wt.% $Ni_2P/TCN$.

**Table 1.** Specific surface area, pore volume, and pore diameter of catalysts.

| Sample | $S_{BET}$ ($m^2 \cdot g^{-1}$) | Total Pore Volume ($cm^3 \cdot g^{-1}$) | Average Pore Diameter (nm) |
|---|---|---|---|
| CN | 74 | 0.39 | 22.74 |
| TCN | 95 | 0.50 | 21.80 |
| 3 wt.% $Ni_2P/TCN$ | 88 | 0.42 | 19.46 |

In order to investigate the surface chemical state of the catalysts, XPS analyses of pure TCN and 3 wt.% $Ni_2P/TCN$ were carried out. As shown in Figure 5a C 1s spectrum, two C 1s peaks are located at 284.8 eV and 288.3 eV, which were attributed to graphitized C and sp2 hybridized C atoms (N=C-N), respectively [38,48]. As shown in Figure 5b N 1s spectra, there were three peaks at 398.8, 400.8, and 404.3 eV, indicating C-N=C, N-(C3), and amino functional groups (-$NH_x$), respectively [49–51]. The C 1s and N 1s spectra of TCN and $Ni_2P/TCN$ had the same binding energy positions, suggesting that the surface chemical state of TCN combined with $Ni_2P$ had no change. As shown in Figure 5c Ni 2p, the peaks at 855.8 eV and 873.5 eV corresponded to $Ni^{2+}$ $2p_{3/2}$ and $Ni^{2+}$ $2p_{1/2}$ spin orbitals, respectively. Compared to metallic Ni, the binding energy of Ni in $Ni_2P$ shifted to higher energies, which was due to the strong electronegativity of P in $Ni_2P$ [43,52]. In addition, the two satellite peaks were located at 861.7 and 880 eV, respectively. As shown in Figure 5d, in P 2p spectrum, the peak at 131.9 eV and 133.4 eV were, respectively, assigned to P $2p_{3/2}$ and P $2p_{1/2}$, indicating that P was negatively charged ($P^{\delta-}$) in $Ni_2P$. The intensities of Ni and P peaks were not very high. However, obvious small particles could be observed through transmission electron microscopy. This phenomenon may be attributed to the fact that $Ni_2P$ was loaded into the interior of the TCN nanotubes, which was consistent with the results obtained from the BET results.

A quantitative ICP analysis of the 3 wt.% $Ni_2P/TCN$ catalyst was carried out, in which the theoretical loading of Ni was 3 wt.%; a quantitative analysis of the actual genus loading in this catalyst revealed that the actual loading of Ni was 2.67 wt.%, which was about 89% of the theoretical value, and the actual loading of P was 0.21 wt.%. The difference between the Ni loading and the theoretical value was assumed to be related to the incorporation of P.

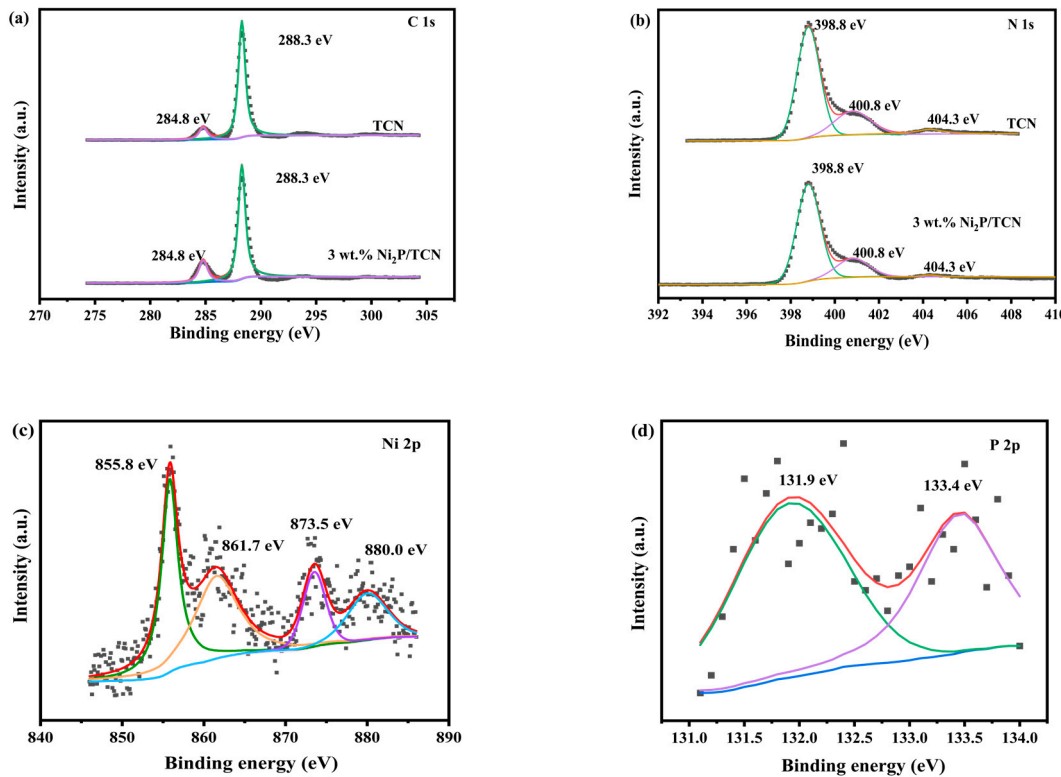

**Figure 5.** XPS spectra of (**a**) C 1s, (**b**) N 1s, (**c**) Ni 2p, (**d**) P 2p for TCN, 3 wt.% Ni$_2$P/TCN.

The light absorption ability was an important factor in evaluating the photocatalytic performance of photocatalysts, so the absorption spectra of the catalysts were analyzed by UV–Vis DRS. The light absorption properties of CN, TCN and 3 wt.% Ni$_2$P/TCN were investigated as shown in Figure 6. The light absorption edges of CN and TCN were both at 439 nm, indicating that they have similar light absorption properties. The 3 wt.% Ni$_2$P/TCN showed strong absorption properties in both the UV and visible ranges, which suggested that the loading of Ni$_2$P improved the photoresponsive ability of the material. The bandgap energies of the samples were tested by Tauc curves, and the bandgap of both CN and TCN was 2.62 eV, indicating that the prepared one-dimensional tubular TCNs had little effect on the energy band structure. Compared with pure TCN, the band gap of the catalyst was reduced to 2.50 eV after Ni$_2$P modification, which indicated that the composite had higher visible light utilization efficiency and significantly enhanced light absorption intensity.

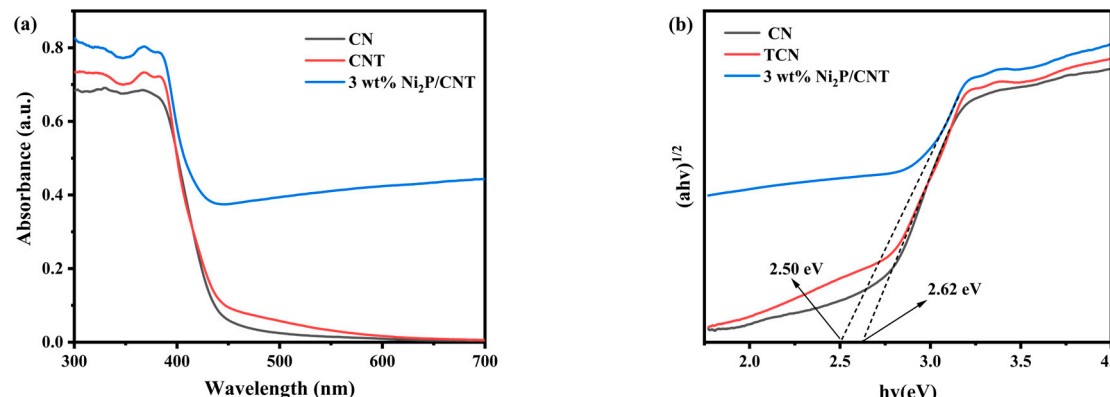

**Figure 6.** (**a**) UV–Vis DRS spectra of CN, TCN, and 3 wt.% Ni$_2$P/TCN; (**b**) Tauc plots of CN and TCN.

The photoluminescence analysis was carried out to assess the photogenerated carrier separation and complexation efficiency. The PL spectra of CN, TCN and $Ni_2P/TCN$ excited by light at 350 nm wavelength showed an intense broad peak at about 442 nm. As could be seen in Figure 7a, the peak intensity of TCN with tubular structure was significantly reduced compared with CN, which was attributed to the fact that TCN with a one-dimensional tubular structure could promote the linear movement of charge, reduce its diffusion distance to the surface, and facilitate the separated migration of photogenerated carriers. In Figure 7a, it could also be seen that the $Ni_2P/TCN$ composites had the smallest fluorescence intensity, which indicated that $Ni_2P$ was well loaded on the surface of TCN and was strongly bonded to it, and the $Ni_2P$ on the surface could capture the electrons on the surface of TCN and promote the migration and separation of photogenerated carriers, thus effectively reducing the complexation rate of photogenerated $e^--h^+$ pairs. This improvement in photogenerated carrier separation efficiency led to an increase in the number of $e^-$ and $h^+$ pairs in the photocatalytic process of $Ni_2P/TCN$. Therefore, the loading of $Ni_2P$ on tubular TCN was potentially valuable for the improvement of photocatalytic hydrogen production from water decomposition. In addition, Figure 7b showed time-resolved PL spectra of TCN catalysts before and after loading $Ni_2P$ co-catalysts, probing the carrier transport of $Ni_2P/TCN$ with TCN. Here, the average emission lifetime of $Ni_2P/TCN$ (2.90 ns) was lower than that of TCN (3.30 ns), and the decrease in fluorescence lifetime suggested a faster charge transfer, which led to a higher carrier separation efficiency [53]. This was consistent with previous test results.

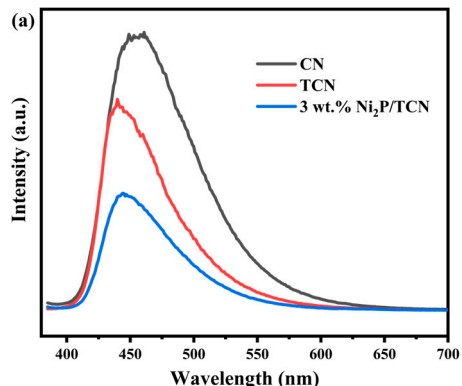 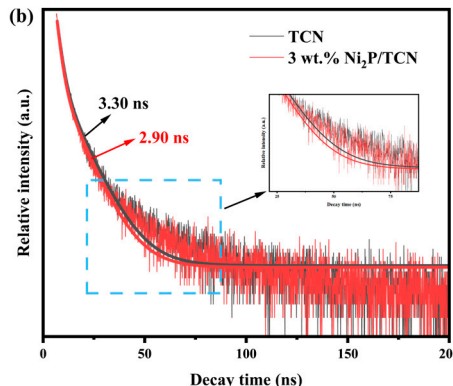

**Figure 7.** (**a**) PL spectra and (**b**) TRPL spectra of samples.

The charge migration efficiency was represented by the EIS spectra as shown in Figure 8a. The EIS radius increased in the order of CN < TCN < 3 wt.% $Ni_2P/TCN$, indicating that the one-dimensional tubular structure in TCN had a smaller charge transfer resistance and the charge migrated to the surface more readily. The 3 wt.% $Ni_2P/TCN$ exhibited the smallest radius, which suggested that the loading of $Ni_2P$ significantly reduced the charge surface migration resistance. The synergistic effect of tubular structure and $Ni_2P$ co-catalyst showed the minimum migration resistance.

From Figure 8b, it could be seen that the magnitude of photocurrent response values was 3 wt.% $Ni_2P/TCN$ > TCN > CN, which implied that the tubular structure had a better visible-light responsiveness and separation of photogenerated electron–hole pairs compared to bulk CN, and the excitation of the photogenerated electrons and the rate of electron mobility were further enhanced with the $Ni_2P$ loading. In summary, the tubular structure and $Ni_2P$ loading shown minimal charge migration impedance and accelerate the carrier separation. Accelerating electron migration from TCN to $Ni_2P$ by constructing an efficient electron transport channel facilitates the transfer of electrons between the catalyst surface and the aqueous phase and improved the catalyst photocatalytic hydrogen production performance.

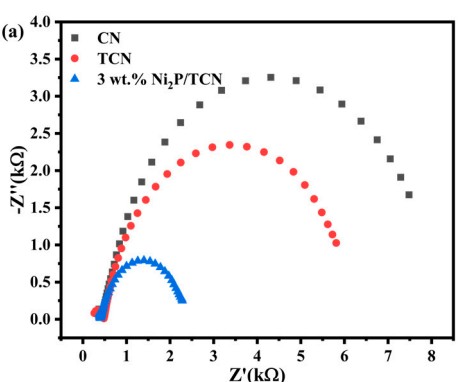
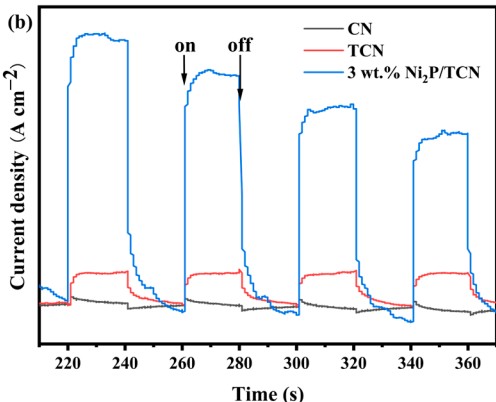

**Figure 8.** (**a**) EIS (**b**) I-t of CN, TCN, and 3 wt.% $Ni_2P/TCN$.

## 2.2. Performance Analysis of Photocatalysts

As shown in Figure 9a,b, the effect of different $Ni_2P$ loadings on the hydrogen production rate of the catalysts under visible light were investigated by adjusting the mass fraction of Ni relative to the support during the catalyst preparation at a fixed molar ratio (Ni:P = 1:10). It could be seen that TCN did not have photocatalytic hydrogen production activity, while P/TCN modified with P alone did not promote the photocatalytic hydrogen production performance. TCN was modified by $Ni_2P$ co-catalysts on the surface, and all the synthesised $Ni_2P/TCN$ catalysts showed remarkable hydrogen production performance. The photocatalytic hydrogen production performance increased with the increase of $Ni_2P$ loading, and the maximum performance was achieved at 3 wt.% Ni loading under Ni:P = 1:10. The 3 wt.% $Ni_2P/TCN$ catalysts had the maximum hydrogen production rate ($3715 \ \mu mol \cdot h^{-1} \cdot g^{-1}$). After determining the 3 wt.% Ni loading, the hydrogen production performance of Ni and P at different molar ratios was further explored, as shown in Figure 9c,d. When the molar ratio of Ni and P was 1:0, the photocatalytic hydrogen production rate was only $302 \ \mu mol \cdot h^{-1} \cdot g^{-1}$, as the molar ratio increased, the photocatalytic hydrogen production performance of the prepared catalyst reaches its maximum when the molar ratio was 1:10, which was 12.3 times higher than that of the 3 wt.% Ni/TCN catalyst. This indicated that P in $Ni_2P$ could alter the electronic structure of clusters, promote the adsorption of $H^+$, and improve the performance of photocatalytic hydrogen production. As the molar ratio further increased, the photocatalytic hydrogen production performance decreased.

The hydrogen production properties of 3 wt.% $Ni_2P/CN$ and 2 wt.% Pt/TCN were also tested, and as shown in Figure 9e, the hydrogen production activity of 3 wt.% $Ni_2P/TCN$ ($3715 \ \mu mol \cdot h^{-1} \cdot g^{-1}$) was much higher than that of 3 wt.% $Ni_2P/CN$ ($870 \ \mu mol \cdot h^{-1} \cdot g^{-1}$) and comparable to the hydrogen production activity of 2 wt.% Pt/TCN ($3672 \ \mu mol \cdot h^{-1} \cdot g^{-1}$), which was attributed to the fact that the hollow nanotube structure had a higher specific surface area compared to CN, thus demonstrating a better light utilization efficiency. Meanwhile, its one-dimensional tubular structure reduced the diffusion distance from the bulk phase to the surface and carriers migrate along the one-dimensional direction, which could accelerate the separation of photogenerated carriers [54]. In summary, the tubular structure of TCN and the loading of $Ni_2P$ improved the light utilization efficiency, facilitated the separation and migration of photogenerated carriers, increased the number of electron–hole pairs, and optimized the adsorption of $H^+$. The combination of them resulted in a significant improvement in the photocatalytic hydrogen production performance of carbon nitride. As shown in Figure 9f, the apparent quantum efficiencies of 3 wt.% $Ni_2P/TCN$ were 3.90% and 1.14% at 380 nm and 420 nm, respectively. As shown in Table 2, the prepared $Ni_2P/TCN$ composites exhibited unrivalled performance advantages under the synergistic effect of the one-dimensional tubular structure and $Ni_2P$. This was attributed to its superb carrier separation efficiency and efficient transport channel for electron migration towards $Ni_2P$.

In order to study the stability of the $Ni_2P$/TCN catalyst, the photocatalytic hydrogen production reaction was carried out for 16 h, as shown in Figure 10. After 16 h of reaction, the 3 wt.% $Ni_2P$/TCN catalyst still showed good stability in the photocatalyst. In addition, it could be seen from the FT-IR spectrum that the internal structure of the catalyst had not changed before and after the reaction.

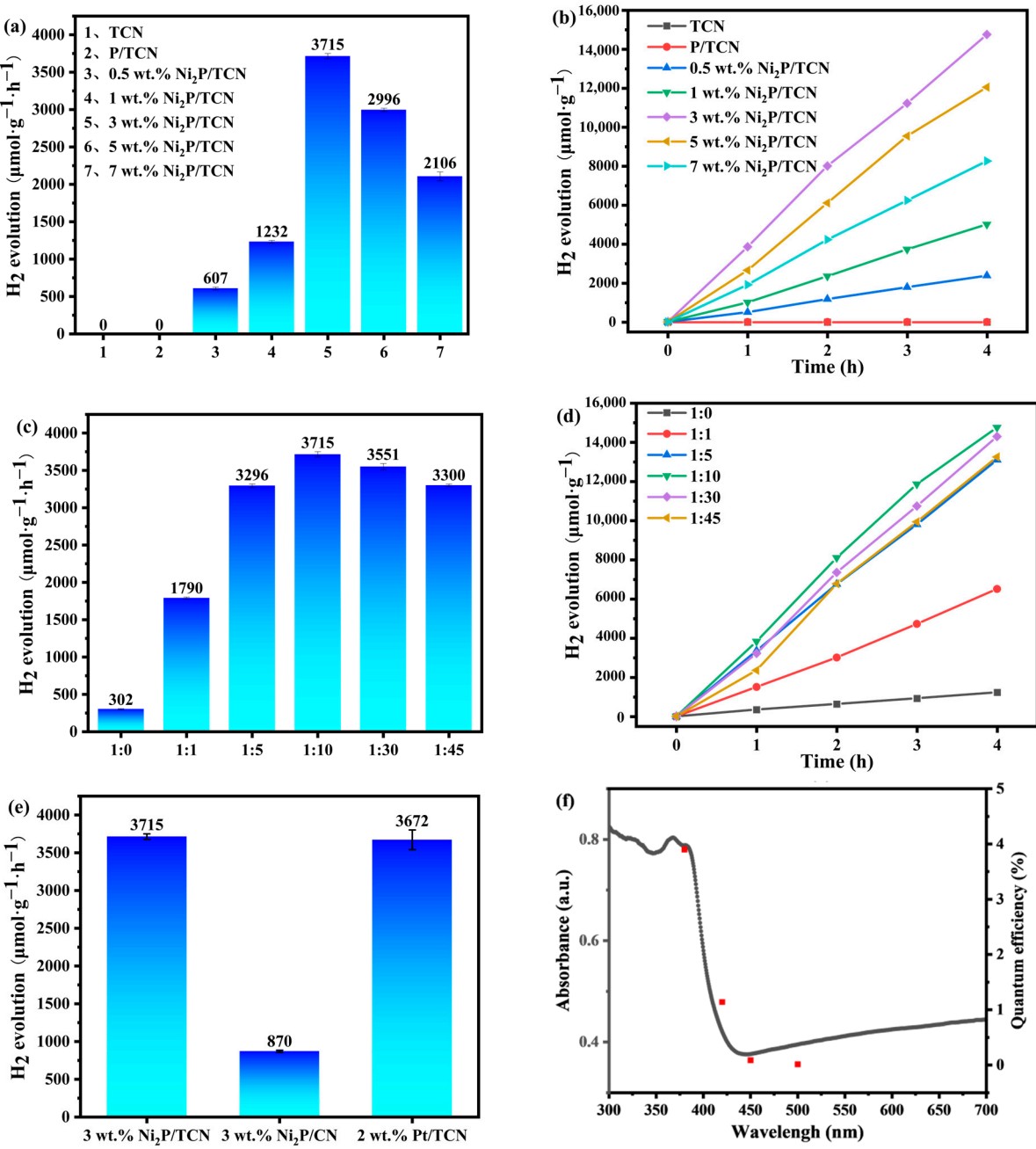

**Figure 9.** (**a**) Hydrogen production performance diagram and (**b**) hydrogen production rate diagram of the catalysts. (**c**) Hydrogen production performance diagram and (**d**) hydrogen production rate diagram of the catalysts with different molar ratios. (**e**) Hydrogen production performance diagram of 3 wt.% $Ni_2P$/TCN, 3 wt.% $Ni_2P$/CN, and 2 wt.% Pt/TCN. (**f**) Apparent quantum efficiency.

**Table 2.** Summary of papers on Ni species and carbon nitride composites in recent years.

| Catalysts | Synthetic Method | Light Source | Activity ($\mu mol \cdot g^{-1} \cdot h^{-1}$) | Reference |
|---|---|---|---|---|
| $Ni/NiO/g\text{-}C_3N_4$ | solvothermal method | 300 W Xe-lamp ($\lambda > 420$ nm) | 2310 | [55] |
| $Ni_xP_y/g\text{-}C_3N_4$ | hydrothermal method | 500 W Xe-lamp ($\lambda > 420$ nm) | 162 | [56] |
| $WO_3/g\text{-}C_3N_4/Ni(OH)_x$ | photodeposition method | 300 W Xe-lamp ($\lambda > 400$ nm) | 576 | [57] |
| $Ni_{12}P_5/g\text{-}C_3N_4$ | mechanical grinding method | 300 W Xe-lamp ($\lambda > 420$ nm) | 126.6 | [58] |
| DCN-Ni | chemical reduction method | 300 W Xe-lamp ($\lambda > 420$ nm) | 449 | [59] |
| CN-0.2Ni-HO | high-temperature hydrogen reduction | 300 W Xe-lamp ($\lambda > 420$ nm) | 354.9 | [60] |
| $Co(OH)_2/g\text{-}C_3N_4/Ni(OH)_2$ | solvothermal method | 300 W Xe-lamp ($\lambda > 420$ nm) | 899 | [61] |
| $NiP/g\text{-}C_3N_4$ | chemical reduction method | 350 W Xe-lamp ($\lambda > 420$ nm) | 1506 | [43] |
| NiS/SO-PCN | solvothermal method | 300 W Xe-lamp ($\lambda > 420$ nm) | 1239 | [62] |
| $Ni_2P/P\text{-}PCN$ | high-temperature phosphating process | 300 W Xe-lamp ($\lambda > 420$ nm) | 1250 | [44] |
| $Ni_2P/TCN$ | photodeposition | 300 W Xe-lamp ($\lambda > 400$ nm) | 3715 | this work |

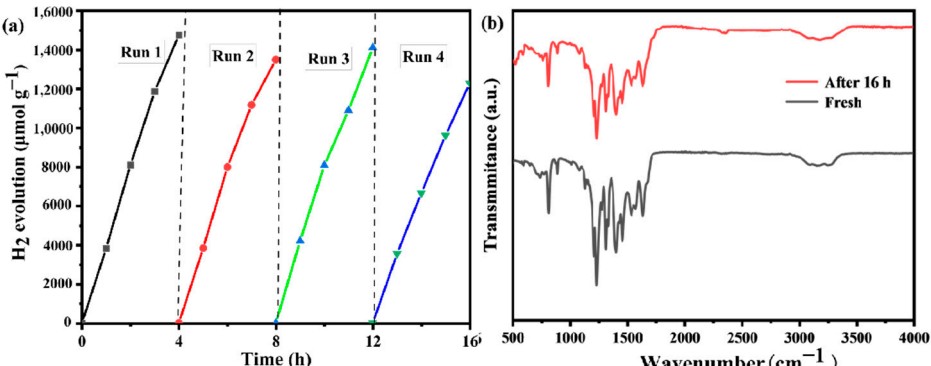

**Figure 10.** (**a**) Hydrogen-producing activity of four cycles of 3 wt.% $Ni_2P/TCN$; (**b**) FT-IR spectra of catalyst before and after cyclic hydrogen production experiments.

### 2.3. Reaction Mechanism Studies

The one-dimensional tubular structure of TCN reduced the diffusion distance of carriers from the bulk to the surface and migrates along a specific direction, which facilitated the separation of electron–hole pairs [54]. In addition, the inner and outer surface area of the tubular structure exposed more active sites, which improved the photocatalytic hydrogen production performance of the catalyst. As shown in Figure 11, the photocatalytic hydrogen production reaction mechanism of $Ni_2P$ loaded onto tubular TCN was as follows: firstly, light was irradiated on the surface of the $Ni_2P/TCN$ catalyst, the photocatalyst absorbed the photons to gain energy, and the electrons in the VB of the TCN were excited to the CB, thus generating electrons and holes. Subsequently, the electrons and holes began to migrate. Due to the loading of $Ni_2P$, the electrons could be easily transferred from TCN to the $Ni_2P$ surface, resulting in the separation of photogenerated carriers [63]. The P atom in $Ni_2P$, due to its strong electronegativity, not only caused electrons from the neighboring Ni atoms to change the charge density distribution but also acted as a basic group to capture $H^+$ and promoted the adsorption of $H^+$, thus enhancing the performance of photocatalytic hydrogen production [64,65]. Meanwhile the hole reached the TCN surface after linear movement on the one-dimensional TCN and was subsequently trapped and neutralized by the sacrificial agent. In conclusion, the light-absorbing ability of the catalyst was improved by combining $Ni_2P$ with TCN, which accelerated the separation of electron–hole pairs, promoted the adsorption of $H^+$, and ultimately facilitated the production of $H_2$.

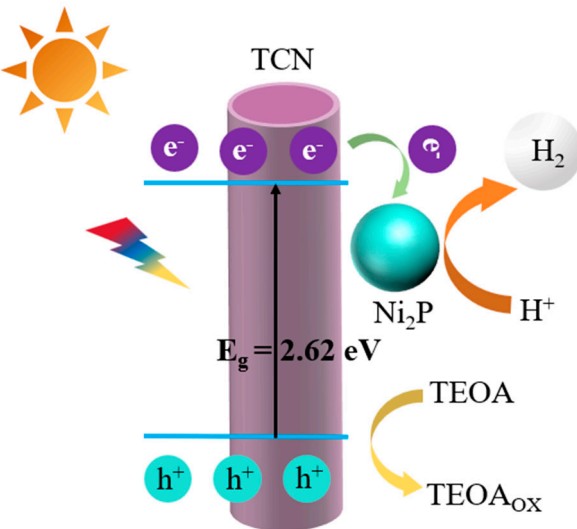

**Figure 11.** Schematic diagram for the photocatalytic H$_2$ generation.

### 3. Experimental Section

#### 3.1. Materials

Melamine (C$_3$H$_6$N$_6$), chloroplatinic acid hexahydrate (H$_2$PtCl$_6$·6H$_2$O), nickel nitrate hexahydrate (Ni(NO$_3$)$_2$·6H$_2$O) and sodium hypophosphite (NaH$_2$PO$_2$·6H$_2$O) were purchased from Shanghai Aladdin Reagent Co., Ltd., Shanghai, China. Triethanolamine (C$_6$H$_{15}$NO$_3$) was purchased from Sinopharm Chemical Reagent Co., Ltd., Shanghai, China. Ethanol (C$_2$H$_6$O) and urea (CH$_4$N$_2$O) were purchased from Wuxi Yasheng Chemical Co., Ltd., Wuxi, China.

#### 3.2. Preparation of CN

Briefly, 5 g of urea was placed in a magnetic boat, sealed into a muffle furnace, heated from 50 °C to 550 °C at a temperature increase rate of 5 °C min$^{-1}$, and then calcined for 4 h. After cooling to room temperature, the yellow powder obtained was named CN.

#### 3.3. Preparation of Tubular TCN

Briefly, 5 g of urea and 0.5 g of melamine were placed in a mortar and ground for 20 min to make a homogeneous mixture. Subsequently, the mixture was placed in a magnetic boat and sealed into a tube furnace, and N$_2$ was introduced; after 10 min, it was heated from 50 °C to 550 °C at a temperature increase rate of 5 °C min$^{-1}$ and calcined for 4 h. After the temperature was cooled down to room temperature, the yellowish powder obtained was named TCN.

#### 3.4. Preparation of Ni$_2$P/TCN

Ni$_2$P/TCN composites were prepared via the photoreduction method. Firstly, 60 mg of TCN powder was dispersed in 40 mL of 10% triethanolamine (TEOA) solution and sonicated for 2 h. NaH$_2$PO$_2$·6H$_2$O (10 mg·mL$^{-1}$) was injected to form a homogeneous solution. Subsequently, a certain volume of Ni(NO$_3$)$_2$·6H$_2$O (10 mg·mL$^{-1}$) solution was injected dropwise into the above system. Then, an oxygen-free atmosphere was formed by bubbling with pure N$_2$ for 20 min. After irradiation under a 300 W Xe lamp (CE Au Light, CEL-HXF300) for 0.5 h, centrifugal washing and filtration were performed to collect the formed deposit, which was dried in a blast-drying oven at 60 °C to obtain the Ni$_2$P/TCN catalyst. The preparation flow chart was shown in Figure 12.

**Figure 12.** Preparation diagram of Ni$_2$P/TCN.

To investigate the effect of different loadings of Ni$_2$P on the photocatalytic performance of Ni$_2$P/TCN catalysts, the molar ratio of added Ni and P was controlled at 1:10, and the loadings of Ni relative to TCN were adjusted to be 0, 0.5, 1, 3, 5 and 7 wt.%. In this study, the corresponding catalysts were labelled α wt.% Ni$_2$P/TCN, where α wt.% denoted the loading of Ni loaded to TCN. The effect of adding excess P on the photocatalytic performance of Ni$_2$P/TCN catalysts was further explored by keeping Ni content at a constant and adjusting the molar ratios of Ni and P to be 1:0, 1:1, 1:5, 1:10, 1:20, 1:30, and 1:45. Finally, the catalyst with Ni loading of 3 wt.% and a molar ratio of Ni to P of 1:10 was named 3 wt.% Ni$_2$P/TCN.

### 3.5. Characterization

X-ray diffraction (XRD) measurements were performed using a SmartLab type X-ray diffractometer at 40 kV and 100 mA using Cu Kα radiation (λ = 1.5406 Å). Fourier transform infrared spectra (FTIR) were obtained on a Nicolet 6700 spectrometer using KBr as a reference. The catalyst microstructure was analyzed using scanning electron microscopy (SEM, TM3000) and transmission electron microscopy (TEM, JEOL-794). Nitrogen adsorption–desorption isotherms were analyzed on an ASAP-type tester. X-ray photoelectron spectroscopy (XPS) was performed and the results analyzed using an ESCALAB250X X-ray photoelectron spectrometer, and measurements were made using a Perkin Elmer HI5000CESCA system with a dual X-ray source featuring 300 W Al Kα radiation and 15 kW of accelerating power. An Agilent-5110 was used to qualitatively and quantitatively analyze the elements in the catalysts. The UV–Vis diffuse reflectance spectra (UV-vis DRS) of all catalysts were tested and analyzed by a UV-3101PC UV—Vis spectrophotometer. The photoluminescence (PL) spectra were measured by Fluoro Max-4 fluorescence spectrometer. The time-resolved photoluminescence (TRPL) were obtained using a FLS980 fluorescence spectrometer (Edinburgh Instruments, Edinburgh, Scotland, UK). Photoelectrochemical tests were performed using a standard three-electrode (Pt, Ag/AgCl and ITO electrodes) system using an electrochemical analyzer (CHI660E). The working electrode was prepared as follows: firstly, 1 mg of catalyst powder was dispersed in a mixture of 5 µL of ethanol, 10 µL of 5% Nafion solution, and 95 µL of deionized water, and after ultrasonication for 60 min, the slurry was coated onto ITO glass and dried to obtain the working electrode, with an electrolyte of 0.5 M Na$_2$SO$_4$ solution.

### 3.6. Photocatalytic Performance

The photocatalytic reaction setup was a closed glass reactor with quartz windows and a volume of 160 cm$^3$. For the photocatalytic reaction, 10 mg of photocatalyst was added to 40 mL of a 10 vol% solution of triethanolamine. The above solution was sonicated for 30 min and then poured into the reactor, and then N$_2$ was passed for 30 min to remove the air from the reactor. The light source was a 300 W Xe lamp (CE Au Light, CEL-HXF300) with a 400 nm cutoff filter for visible light irradiation of the photocatalytic reactor. The sampled gases were analyzed qualitatively and quantitatively by gas chromatography (GC-7860 Plus) using high-purity nitrogen as carrier gas.

### 4. Conclusions

In conclusion, nickel phosphide (Ni$_2$P) was loaded onto one-dimensional tubular graphitic carbon nitride (TCN) using a facile high-temperature thermal polymerization

and photodeposition method. The tubular structure of TCN provided a larger specific surface area, in addition to which the tubular one-dimensional structure shortened the diffusion distance from the bulk phase to the surface and induced a one-dimensional linear migration of carriers, which inhibited the complexation of photogenerated carriers and increased the number of electron–hole pairs. The $Ni_2P$ was uniformly dispersed on the TCN surface in the form of nanoclusters, which accelerated the electron migration from TCN to $Ni_2P$ by constructing efficient electron transport channels and promoted the charge separation efficiency. In addition, the P atoms in $Ni_2P$ could act as basic groups to capture $H^+$ due to their strong electronegativity. Under the synergistic effect of the one-dimensional tubular structure and the $Ni_2P$ co-catalyst, the 3 wt.% $Ni_2P$/TCN had the best hydrogen production performance ($3715\ \mu mol \cdot h^{-1} \cdot g^{-1}$) and exhibited good stability. This work provides a new method for the further development of $g$-$C_3N_4$-based photocatalytic hydrogen precipitation catalysts.

**Author Contributions:** Investigation, Data curation, Writing—Original draft preparation, C.J.; Software, Y.J.; Investigation, F.L.; Writing—Reviewing and Editing, C.F.; Writing—Reviewing and Editing, Supervision, Project administration, J.D.; Validation, Resources, H.W.; Writing—Reviewing and Editing, P.Z.; Conceptualization, Resources, Project administration, Funding acquisition, G.G. All authors have read and agreed to the published version of the manuscript.

**Funding:** This research was funded by the National Natural Science Foundation of China (No. 21706131, 21878159, 22078159 and 22278213).

**Data Availability Statement:** The authors confirm that the data supporting the findings of this study are available within the article.

**Conflicts of Interest:** The authors declare no conflict of interest.

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
