# Peer review of "One-Dimensional Tubular Carbon Nitride Embedded in Ni2P for Enhanced Photocatalytic Activity of H2 Evolution"

_catalysts, doi:10.3390/catal14040243_

Round 1
Reviewer 1 Report
Comments and Suggestions for Authors The manuscript “One-dimensional tubular carbon nitride embedded by NiP for enhanced photocatalytic activity of H2 evolution” by Ch. Jiang et all. belongs to important investigations clearly describing effect of CNT inner and outer surface modification on photocatalytic activity in H2 evolution. This study was carried out on a good experimental level. The authors received interesting results concerning the fundamental process of a charge transfer in a complicate catalytic system. However, the text of this article contains many inaccuracies and negligence.- Line 97. Using or with?
- Line 101. NiP should be instead of P.
- Lines 108-109. “while effectively accelerating the kinetics of the photocatalytic reaction by promoting mass transfer” – promoting the photogenerated carriers transfer.
- Line 154. What means ICP abbreviation?
- Lines 156-157. “a quantitative analysis of the actual genus loading in this catalyst revealed that the actual loading of Ni was 2.86 wt.%, which was about 95 % of the theoretical value, and the actual loading of P was 0.08 wt.%.???
As follows from atomic weight, (Ni – 58.7 and P – 31), the sample containing 3 wt% of nickel phosphide contains 1.96 wt% Ni and 1.04 wt% P. Something was wrong with analysis. In addition, the sensitivity for P was too low.
- Line 165. Why Taus? It should be Tauc.The Tauc method is the conventional technique for estimating the accurate optical bandgap energy of semiconductors using UV–Vis spectroscopy.
- From Line 188. There is no correspondence between the Figure 7b and its description. The curves shown in the figure are visually completely identical.
- Line 217. Up to this point, modification with individual phosphorus was not mentioned.
- Line 221. If authors speak about modification by NiP compound, Again, and if NiP loading is 3 wt%, the pure Ni loading is ~2 wt% (see comment 5).
- Lines 224-228. There is no phosphorous in Ni/TCN catalyst. The phrase from Line 224 to Line 228 should be revised. If authors describe effect of TCN modification by NiP compound, Ni:P atomic ratio could be only 1:1 (as in Figs 9 a and b). The description of the TCN modification effect by NiP and by the individual components Ni and P should be separated. It is necessary to show why modification with the compound is more promising.
- Line 252. Table 2, not 5-2.
- The “Reaction mechanism” seems to be a combination of experimental data and ideas from literature. Thus, references are needed.
- Line 325. There is significant description between "Preparation of NiP/TCN" (Lines 325-326) and the samples characterization. As follows from the “Preparation...”, the catalyst contained 3 wt% Ni and its formula was NiP10. This catalyst was named as 3 wt% NiP/CN (or TCN??). However, as follows from Fig. 3j, it was the nickel phosphide (NiP) that was detected on the TCN surface. If all experiments were performed on samples containing excess phosphorus, THIS MUST BE SPECIFICALLY NOTED.
- Line 373. According to Figure 9e and Table 2, the catalyst of 3 wt% NiP/TCN has the best catalytic performance. 3 wt% NiP/CN is the worst (870 μmol·h−1·g−1).
Reviewer 2 Report
Comments and Suggestions for Authors
This work reports the reparation of nickel phosphide supported on one-dimensional tubular carbon nitride. The results are of interest and a good yield of hydrogen production is reported. However, some issues need to be addressed before the publication:
1. There is a missing sentence in the introduction, as the authors indicate that this NiP-CN system has already been studied in photocatalytic hydrogen production. Therefore, the contribution of this work, i.e. the tubular form, should be highlighted.
2. Although XPS results seem to match well with the other results, they must be confirmed, since some of the signals shown do not fit the envelopes, particularly in Ni 1s and P 2p.
3. There are some inconsistencies in the analysis of the results, since in some of them the authors suggest that the NiP could be inside the TCN tubes, and in another they state that the NiP was well loaded on the surface of the TCN. In the latest case, why is the NiP not well observed by XPS?
4. In the mechanism, is there any evidence of the basic nature of the NiP species?
5. Correct some typos (lines 165 and 252 i.e.).
Maybe the main drawback of this work is the lack of analysis and discussion of some of the results. Then, I do not recommend this manuscript for publication in the present conditions.
Round 2
Reviewer 2 Report
Comments and Suggestions for Authors
No more comments